# Comprehensive Evaluation of Toxicological Profile and Immunomodulatory Impact of an Immune Enhancing Oral Liquid

**DOI:** 10.3390/foods14183166

**Published:** 2025-09-11

**Authors:** Hongzhang Chen, Zhuming Ye, Sihe Chen, Wenquan Huang, Ying Lin, Qixin Kan

**Affiliations:** 1Guangdong Key Laboratory of Fermentation and Enzyme Engineering, School of Bio and Chemical Engineering, South China University of Technology, Guangzhou 510006, China; hongzhang.chen@infinitus-int.com; 2Guangdong Research Center of Industrial Enzyme and Green Manufacturing Technology, School of Bio and Chemical Engineering, South China University of Technology, Guangzhou 510006, China; 3Guangdong Provincial Key Laboratory of Nutraceuticals and Functional Foods, College of Food Sciences, South China Agricultural University, Guangzhou 510642, China; louisyzm@outlook.com (Z.Y.);

**Keywords:** immune enhancing oral liquid, toxicological profile, immunomodulatory efficacy, *Polygonum multiflorum* Thunb., *Morinda officinalis* How

## Abstract

Traditional Chinese herbal medicines are characterized by multi-component synergy and have garnered significant attention in the healthcare research. The Immune Enhancing Oral Liquid, formulated with *Polygonum multiflorum* Thunb. and *Morinda officinalis* How extracts, has demonstrated biological activities including gut microbiota modulation. To systematically evaluate its safety profile, toxicity, and immune-enhancing efficacy, this study assessed toxicological and immunomodulatory parameters in murine models through the following protocols: acute toxicity testing; genotoxicity assays; 28-day oral toxicity study; and immunological function assessments encompassing cellular immunity, humoral immunity, non-specific immunity, and cytotoxic function. Results indicated that the immune enhancing oral liquid showed no obvious acute toxicity, genotoxicity, or 28-day oral toxicity. Furthermore, the immune enhancing oral liquid exhibited no adverse effects on murine hematopoietic or hepatic functions. Crucially, the immune enhancing oral liquid significantly enhanced immune responses, as evidenced by promoted lymphocyte proliferation, elevated serum hemolysin levels, augmented macrophage phagocytic activity, and heightened NK cell activity. Immune enhancing oral liquid was recognized as safe and non-toxic, and the study provides a scientific basis for the application of the oral liquid, validating the effectiveness of the combination of traditional herbal wisdom and modern pharmacology.

## 1. Introduction

Traditional Chinese herbal medicine (TCHM), characterized by its multi-component synergism, systemic low toxicity, and adaptability to chronic disease management, has regained prominence within global nutraceutical systems [1,2]. In Asian countries such as China, Japan, and Korea, TCHM formulations have been employed for millennia in disease management, health maintenance, and life extension [3]. These classical formulations are extensively utilized in foods, beverages, soups, and pharmaceuticals, enjoying unparalleled cultural and therapeutic recognition across East Asia [4,5,6]. China’s medical canon encompasses over 12,000 classical texts documenting more than 10,000 herbal formulations, with “medicinal and edible homology” herbs serving as focal points for health-oriented nutraceutical development [7].

The third edition of China’s Medicinal and Edible Homology Catalogue, jointly issued by the National Health Commission and State Administration for Market Regulation, lists 102 approved substances, including *Morinda officinalis* How, Codonopsis pilosula, Cistanche deserticola, Dendrobium officinale, Panax quinquefolius, Astragalus membranaceus, Ganoderma lucidum, Cornus officinalis, Gastrodia elata, and Eucommia ulmoides leaf [8]. Among these, *Polygonum multiflorum* Thunb. (PM), a widely used tonic herb, has demonstrated multifaceted health benefits. Phytochemical analysis has revealed that its stilbenes and polyphenols confer anti-aging effects by inhibiting oxidative stress and enhancing mitochondrial function. Moreover, its polysaccharides and anthraquinone glycosides modulate T and B lymphocyte activity, thereby enhancing immune responses and reinforcing immune defenses [9]. *Morinda officinalis* How (MO), a plant of the Rubiaceae family, contains bioactive polysaccharides that have been shown to significantly increase murine macrophage phagocytosis, thereby potentiating innate immunity [10]. The polysaccharide from MO promotes the expression of monocyte chemoattractant protein-1 and interleukin-8 in the cells of hepatic stellate cell slices from beagle dogs, thereby regulating immune function [11]. At a molecular level, polysaccharides from herbs like PM are reported to exert immunomodulatory effects by activating macrophages and dendritic cells via pattern recognition receptors, leading to the production of cytokines such as TNF-α, IL-1β, and IL-6, and enhancing phagocytosis [12]. They can also promote lymphocyte proliferation and modulate the Th1/Th2 balance. Compounds in MO, such as polyphenols and terpenoids, may contribute by modulating inflammatory pathways and exhibiting antioxidant activity, which can support immune cell function [10].

Oral liquid, as a form of Chinese herbal medicine formulation, is primarily developed based on the traditional water extraction process, with a small amount of ingredients compounded using supercritical or alcohol extraction methods [3]. A traditional water decoction was employed to mirror the preparation method most commonly used in traditional practice. This method effectively extracts polar compounds such as polysaccharides, tannins, and glycosides. The traditional water extraction process retains a large number of plant polysaccharides and water-soluble polyphenols. Among these, edible botanical polysaccharides have been a promising class of natural compounds for the prevention and management of chronic diseases, positioning them as excellent candidate ingredients for nutraceutical and dietary supplement formulations [6]. However, multiple challenges—including the material diversity of herbal formulas, uncertain component interactions, complex quality control and standardization processes, and unclear mechanisms of action and toxicity—collectively impact the final product quality. Among, safety concerns persist regarding *Polygonum multiflorum* Thunb. hepatotoxicity, attributed to anthraquinones (e.g., emodin) and stilbene glycosides [13]. Pharmacokinetic studies emphasize dose-dependent hepatocyte apoptosis and cholestatic liver injury, necessitating rigorous quality control in product development [14]. Globally, regulatory frameworks for herbal products transcend conventional drug/food paradigms [2]. Compliance with Organisation for Economic Cooperation and Development (OECD) guidelines, Food and Drug Administration (FDA) regulations, and National Medical Products Administration (NMPA) requirements mandates comprehensive toxicological profiling, including acute/chronic toxicity, genotoxicity, reproductive/carcinogenic risks, and immunotoxicity assessments [15]. These evaluations integrate in vivo models, in vitro assays, computational toxicology, and molecular profiling to ensure safety and efficacy [1,15]. The immune enhancing oral liquid incorporates the classical compatibility of PM and MO, formulated in strict accordance with the principles outlined in Records of Syndrome Differentiation. It is designed to tonify the kidneys, strengthen tendons and bones, invigorate the spleen, and eliminate dampness. Additionally, the combined use of PM and MO is also documented in the Zishen Yutai Pills, which adheres to the current standards set by National Medical Products Administration. The combination of PM and MO was selected based on their complementary roles in traditional medicine for nourishing vitality and calming the mind. Synergistically, PM’s purported hematinic and immunostimulatory properties are theorized to be balanced and enhanced by MO’s adaptogenic and anti-inflammatory effects, potentially leading to a more comprehensive modulation of immune function and stress response, while also mitigating potential hepatotoxicity associated with PM alone through MO’s purported hepatoprotective compounds. These pills are principally efficacious in replenishing kidney, fortifying the spleen, augmenting vital energy, nourishing blood to stabilize pregnancy, and enhancing overall physical robustness [16].

Immune enhancing oral liquid, formulated with PM and MO extracts, has demonstrated preliminary modulation of gut microbiota composition and metabolic pathways [17]. This study comprehensively evaluates its toxicological and immunomodulatory profiles through acute toxicity test; genotoxicity assays; 28-day oral toxicity study; immunological parameters: detection of cellular immune function; detection of the humoral immune function; detection of Nonspecific Immune Function; detection of the immune-cell-killing function. Findings will establish scientific foundations for health food, ensuring therapeutic efficacy and consumer safety. This investigation bridges traditional herbal wisdom with modern pharmacological validation, addressing critical gaps in evidence-based traditional herbal medicine applications.

## 2. Materials and Methods

### 2.1. Material and Reagents

*Polygonum multiflorum* Thunb. (PM) and *Morinda officinalis* How (MO) were provided by Infinitus (China) Company Ltd. (Guangdong, China). Paxone, sodium azide, diamino, 1,8-dihydroxyanthraquinone, cyclophosphamide were purchased from Maclin Biochemical Technology Co., Ltd. (Guangdong, China). Histidine-deficient Salmonella Typhi TA97, TA98 (frameshift mutant), TA100 and TA102 (base replacement mutant) strains were purchased from Shanghai Baolu Biotechnology. (Shanghai, China). Other reagents were shown in Appendix B (Table A2).

### 2.2. Animal Experiments

Specific Pathogen Free (SPF) grade mice (weighing 12–22 g) were obtained commercially from Zhejiang Weitong Lihua Experimental Animal Technology Co., Ltd. (Beijing, China). The mice were housed in the Experimental Animal Center of South China Agricultural University under barrier conditions, with temperature maintained at 22–25 °C, relative humidity at 40–60%, and a 12 h light/12 h dark cycle. After a 7-day acclimation period, the mice were randomly assigned into experimental groups, with 10 mice per group. Each cage housed five mice, and total number of animals was 140 mice (female 60, male 80). All animal procedures were conducted in compliance with the approved animal use protocol (SYXK (Yue) 2019-0136) issued by the Animal Ethics Committee of South China Agricultural University. This study also adhered to the internationally recognized US guidelines for the care and use of laboratory animals (NIH Publication No. 85-23, revised in 1985).

### 2.3. Preparation and Analysis of the Immune-Enhancing Oral Liquid

According to the protocol of literature [17], compound oral liquid is based on PM and MO water extracts. The extraction process crushed PM and MO to 10 kg for extraction, and equal mass dried PM and MO were extracted twice with boiling water (1:10, *w*/*v*) for 2 h. Then the extracts were mixed and concentrated to give a compound oral solution. The analysis of the Immune Enhancing Oral Liquid was performed using HPLC and is described in our previous study [17] with the Appendix A and Appendix B (Figure A1 and Table A1). The Immune-Enhancing Oral Liquid was characterized by a highly consistent chemical profile across manufacturing batches, achieved through a stringent quality control regime. This batch-to-batch uniformity in the composition and concentration of active constituents ensures reliable safety and reproducible therapeutic efficacy in clinical applications.

### 2.4. Toxicology Evaluation of the Health-Enhancing Oral Liquid

#### 2.4.1. Acute Toxicity Test

Based on previous literature and with appropriate revisions [15], blank group and sample group with 10 mice each were set in the study and total number of animals was 20 male mice. A standard dose of 15 mL/kg/day was used in all groups. Distilled water was used for sample dissolution in the sample group, and distilled water was used in the blank group. Fasting for 12 h before administration, water, the general condition, activity status, excretion, poisoning symptoms, initiation time, severity, duration, time of death and death of each group were recorded in detail, and observed continuously for 14 days.

#### 2.4.2. Genotoxicity Assays

Three standard genotoxicity tests were used for the testing of immune-enhanced oral fluids, Bacterial reverse mutation assay (the Ames test), Mouse micronucleus test, Mouse sperm malformation experiments. All genotoxicity assays were performed under good laboratory practice conditions.

Bacterial reverse mutation assay (Ames test): To evaluate mutagenic potential, a bacterial reverse mutation assay was conducted following a modified protocol as previously described [18,19]. Salmonella typhimurium strains TA97 and TA98 were employed to detect frameshift mutations, while TA100 and TA102 were used to assess base-pair substitution mutations. Test groups were treated with the oral liquid at doses of 5 mg, 2.5 mg, 1.25 mg, and 0.625 mg per plate. Negative controls received distilled water, while positive controls included: Non-activated system: TA97, TA98, and TA102 treated with Dexon (50.0 µg/plate); TA100 treated with sodium azide (1.5 µg/plate). Metabolically activated system (+S9): TA97, TA98, and TA100 treated with 2-aminofluorene (10.0 µg/plate); TA102 treated with 1,8-dihydroxyanthraquinone (50.0 µg/plate). All strains were inoculated in nutrient broth and incubated at 37 °C for 14 h to achieve a bacterial density of 2 × 108 colony-forming units (CFU)/mL. Test substances were subsequently introduced, and revertant colonies were quantified after 48 h of incubation.

Mouse micronucleus test: set sample group, negative controls and positive controls with 10 mice in the study and total number of animals was 30 mice (famale). The experimental samples included the 20 mL/kg/day oral liquid; negative controls received distilled water; positive controls received 40 mg/kg cyclophosphamide. The test substance was administered via oral gavage once daily for four consecutive days. On the fifth day, bone marrow was harvested from the femurs, and smears were prepared according to standard protocols [20]. The bone marrow cells were flushed from the femurs using fetal bovine serum (FBS) and centrifuged at 1000 rpm for 5 min. The supernatant was discarded, and the cell pellet was resuspended in a small volume of FBS. A drop of the cell suspension was placed on a clean glass slide, and a smear was prepared using a spreader slide at a 45° angle. The smears were air-dried, fixed in absolute methanol for 10 min, and stained with 5% Giemsa solution (diluted in phosphate buffer, pH 6.8) for 15 min. After staining, the slides were gently rinsed with distilled water, air-dried, and mounted with a coverslip using neutral balsam. For each animal, 2000 polychromatic erythrocytes (PCEs) were examined under a light microscope (1000× oil immersion). Micronuclei (MCN) were identified according to the following criteria: (1) round or oval-shaped structures with sharp boundaries; (2) staining intensity similar to that of the main nucleus; (3) diameter between 1/20 and 1/5 of the main nucleus; (4) location within the cytoplasm of the PCE and clearly separated from the main nucleus. The ratio of PCEs to normochromatic erythrocytes (NCEs) was also recorded to assess bone marrow proliferation activity.

Mouse sperm malformation experiments: A sample group, negative control group, and positive control group were established, with 10 male mice per group and total number of animals was 30 mice (male). The experimental group received the oral liquid at a dose of 20 mL/kg/day. The negative control group received distilled water, and the positive control group received 40 mg/kg cyclophosphamide. The test substance was administered via oral gavage once daily for five consecutive days. On the 35th day after the initial treatment, the mice were euthanized by cervical dislocation. Both cauda epididymides were dissected and placed in a Petri dish containing 2 mL of normal saline. The epididymal tissue was minced with scissors to release sperm, and the suspension was filtered through a 100 μm nylon mesh to remove tissue fragments. The sperm suspension was centrifuged at 1000 rpm for 5 min, and the pellet was resuspended in a small volume of saline. A drop of the suspension was smeared onto a clean glass slide and air-dried. Smears were fixed in methanol for 10 min and stained with 2% eosin Y aqueous solution for 1 h. Alternatively, slides were stained with 5% Giemsa solution for 20 min. After staining, the slides were rinsed gently with distilled water, air-dried, and mounted with a coverslip. For each animal, at least 1000 sperm were examined under a light microscope. Sperm abnormalities were classified into the following categories: (1) amorphous heads; (2) banana-shaped heads; (3) hook-less heads; (4) double or multiple heads; (5) folded heads; (6) broken heads; (7) coiled tails; (8) double tails. A spermatozoon was considered malformed if any part of its morphology deviated from the normal hook-shaped head and long, single tail [21].

#### 2.4.3. 28 Day Repeat Oral Toxicity Study

According to the Organization for Economic Cooperation and Development (OECD) guidelines NO.407, some experimental details were adjusted. Blank group and sample group were set in the study. The sample group received the test material via oral gavage at a dose of 20 mL/kg/day in all groups. Total number of animals was 20 mice (famale 10, male 10). The blank control group was maintained on a standard basal diet with distilled water. Following the 28 day repeat oral toxicity study, all animals were fasted overnight. Blood samples for hematological and biochemical analyses were collected. Blood was collected from the retro-orbital plexus into EDTA tubes for hematology and serum separator tubes for clinical biochemistry. Referring to the research methods reported in the literature [22], approximately 1.5 mL of blood was collected from each animal. In the hematological analysis, a hematology analyzer (Siemens, Munich, Germany) was used to measure hemoglobin, red blood cell count, white blood cell count, percentage of neutrophils, neutrophil count, percentage of lymphocytes, lymphocyte count, percentage of monocytes, monocyte count, percentage of eosinophils, eosinophil count, percentage of basophils, basophil count, and platelet count. For serum biochemical analysis, the following were measured: glucose, blood urea nitrogen, creatinine, alanine aminotransferase, aspartate aminotransferase, total protein, albumin, gamma globulin, total cholesterol, and triglycerides.

### 2.5. Effects on Murine Immunity

In accordance with the experimental mice and the culture environment described above, healthy mice with similar body weights and ages were selected and randomly divided into a blank group, a low dose group (1.25 mL/kg), a medium dose group (2.5 mL/kg), and a high dose group (5 mL/kg), with 10 mice in each group and total number of animals was 40 mice (female 20, male 20). Mice in each experimental group were administered the corresponding oral liquid, while the blank group was given an equal volume of water. The experiment lasted for 28 days. After the last administration, the spleens were dissected to measure the spleen to body weight ratio.

#### 2.5.1. Detection of Cellular Immune Function

Splenic Lymphocyte Proliferation Assay: The method was referred to the literature [23] and appropriately modified. One hour after the last administration, the mice were sacrificed by cervical dislocation. The spleens were aseptically excised and placed in a sterile Petri dish containing RPMI 1640 medium. Glass beads were added to the Petri dish and gently shaken to disperse the spleen tissue through the mechanical action of the glass beads. The dispersed cell suspension was filtered through a 70-μm cell strainer to obtain a single cell suspension. The single-cell suspension was transferred to a centrifuge tube and centrifuged at 1000 rpm for 5 min. The supernatant was discarded, and the cells were resuspended in RPMI 1640 medium supplemented with 10% fetal bovine serum. The cell concentration was adjusted to 2.1 × 10^6^ cells/mL. The cell suspension was inoculated into 24 well plates in 1 mL aliquots. In the experimental wells, 5 μg/mL of concanavalin A (ConA) was added to stimulate T cell mitosis, while in the control wells, an equal volume of RPMI 1640 medium without ConA was added. The 24-well plates were incubated in a cell incubator at 37 °C with 5% CO_2_ for 68 h. Four hours before the end of the incubation, 20 μL of MTT solution (5 mg/mL) was added to each well, and the incubation was continued for another 4 h. After the incubation, the supernatant in each well was carefully aspirated, and an appropriate amount of DMSO was added to dissolve the formazan crystals. The optical density of each well was measured at a wavelength of 570 nm using a microplate reader to quantitatively determine lymphocyte proliferation.

Delayed-Type Hypersensitivity (DTH) Response: Consistent with the above grouping, during the feeding process of mice in the blank and sample groups, all mice received a unified subcutaneous injection of 200 μL of 2% sheep red blood cells for sensitization to ensure their good health. Four days after sensitization, 20 μL of 20% sheep red blood cells were injected subcutaneously into the plantar part of the left posterior foot. The paw thickness was measured at 24 and 48 h after injection to observe the changes in the left posterior foot, including redness, swelling, swelling range, and skin temperature.

#### 2.5.2. Detection of the Humoral Immune Function

Serum Hemolysin Assay: The serum hemolysin sheep red blood cell (SRBC) method described in the literature [23] was adopted. Consistent with the above grouping, mice were uniformly administered by gavage. Each group was sensitized with 0.2 mL of 2% SRBC. Four days after immunization, the mice were sacrificed, and the blood was centrifuged at 3000 r/min for 10 min. The serum was collected and diluted 500 fold. Then, 1 mL of the diluted serum, 0.5 mL of 10% SRBC, and 1 mL of 10% complement were added to a test tube. An equal volume of normal saline was used as a blank control. The mixture was shaken, placed in a 37 °C water bath for 30 min, and then ice bathed. After centrifugation at 2000 r/min for 10 min, the supernatant was taken. The value of sheep red blood cell hemolysis (HC50) was measured using a microplate reader at a wavelength of 540 nm.

Antibody-Secreting Cell Quantification: Splenocytes were prepared as previously described according to the grouping and adjusted to a concentration of 5 × 10^6^ cells/mL. For plaque determination, 1% agarose was melted and kept in a 45 °C water bath. It was then mixed with an equal volume of Hanks balanced salt solution and dispensed into small tubes, 1 mL per tube. Subsequently, 100 μL of 10% SRBC and 40 μL of the spleen cell suspension were added to the tube, quickly mixed, and poured onto the surface of the bottom-layer gel to form the top layer gel. After the agar solidified, it was incubated at 37 °C for 1 to 1.5 h. Then, 1 mL of a 10 fold diluted complement solution was added, and the incubation was continued for another 1 to 1.5 h. The number of plaque-forming units was counted and expressed as the number of plaques per 106 spleen cells.

#### 2.5.3. Detection of Nonspecific Immune Function

Carbon Clearance Kinetics: According to the method described in the literature, 10 mice were randomly selected from each group. Thirty minutes after the last administration, a mixture of Indian ink and normal saline (2:1) was injected into the tail vein at a dose of 0.1 mL/10 g/day. Twenty microliters of blood was collected and added to 2 mL of 0.1% Na_2_CO_3_ solution. The absorbance was measured at 640 nm using a spectrophotometer. After blood collection, the mice were sacrificed, and the liver and spleen were weighed.

Peritoneal Macrophage Phagocytosis: Ten mice were randomly selected from each group. One day after the last administration, each mouse was intraperitoneally injected with 1 mL of 20% fresh chicken red blood cell suspension and then sacrificed by cervical dislocation. Two milliliters of normal saline was injected into both sides of the abdominal wall. The normal saline was thoroughly mixed with the peritoneal fluid. One milliliter of the peritoneal fluid was dropped onto a glass slide covered with wet gauze and incubated in a 37 °C incubator for 30 min. Then, it was rinsed with normal saline and dried. The slide was fixed with a 1:1 (*v*/*v*) methanol-acetone solution and stained with a 4% (*v*/*v*) giemsa phosphate buffer for 30 min. The number of phagocytic cells was counted under an oil immersion microscope to calculate the phagocytosis percentage and phagocytosis index.

#### 2.5.4. Detection of the Immune-Cell-Killing Function

Referring to the method described in the literature [23] and making appropriate modifications, the grouping was the same as above. On the 2nd day after the last dose, 0.2 mL of blood was collected from the inner canthus venous plexus of the mice and added to PBS containing heparin. NK cells were separated using a silica colloidal suspension (Percoll) and washed twice with Hanks solution. The cell concentration was adjusted to 1 × 10^7^ cells/mL with RPMI 1640 medium containing calf serum. K562 cells in the logarithmic growth phase were cultured for 24 h, washed with RPMI 1640 medium containing calf serum, and the cell concentration was adjusted to 1 × 10^5^ cells/mL. In a 96 well plate, 100 μL of K562 cells was added to each well, followed by 100 μL of NK cells. In the natural release group, RPMI 1640 medium containing calf serum was used instead of NK cells, and in the maximum release group, 1% NP40 was used instead of NK cells. Three replicate wells were set for each sample. After incubation at 37 °C with 5% CO_2_ for 4 h and centrifugation, the supernatant was taken. The lactate dehydrogenase (LDH) release rate was measured at 570 nm using an LDH kit (Nanjing Jiancheng, Nanjing China).

### 2.6. Statistical Analysis

Data are presented as mean ± standard deviation (SD). Statistical analysis was performed using SPSS 19.0 software. Normality of distribution was assessed using the Shapiro–Wilk test, and homogeneity of variances was verified with Levene’s test. For data meeting parametric assumptions, *t*-test or one-way analysis of variance (ANOVA) was used for comparisons across multiple groups, followed by Dunnett’s post hoc test for comparisons against the control group. For non-parametric data, the Kruskal–Wallis test followed by Dunn’s post hoc test was employed. Dose–response trends were analyzed using linear regression. A value of *p* < 0.05 and *p* < 0.01 were considered statistically significant.

## 3. Results and Discussion

### 3.1. Toxicology Evaluation of Immune Enhancing Oral Liquid

#### 3.1.1. The Analysis of Acute Toxicity Test

The acute toxicity of the compound was evaluated by determining the median lethal dose (LD_50_), which can effectively determine the risk class and grade of compounds [24,25]. In this study, mice were administered the extract via oral gavage at a dose of 15 g/kg/day. During the 14 day observation period, no abnormal mortality, behavioral changes, excretory irregularities, or overt toxic symptoms were observed in either the treatment or control groups. Additionally, upon inspection, there were no obvious pathological changes in the color and texture of important organs such as the liver, kidneys, brain, heart, lungs, spleen, and thymus. According to Consumer Product Safety Commission for acute oral toxicity classification, the LD_50_ of the test formulation under 5 g/kg/day, classifying it as toxic under national safety standards [15]. Compared with previously reported cases of liver toxicity associated with PM extracts, the experiment group mice in the liver and no obvious lesions. According to literature reports, the main toxic components in the water extract of PM are stilbenes, and in the alcohol extract are quinones. The liver toxicity of PM can be reduced during the drying process of steaming and sun drying. In addition, compatible detoxification is one of the traditional Chinese medicine detoxification methods, which is one Chinese medicine is used together with another or several other Chinese medicines to achieve the purpose of inhibiting or eliminating the toxicity of Chinese medicines [26].

#### 3.1.2. The Analysis of Genotoxicity Assays

Bacterial reverse mutation assay: the Ames test, a widely validated as utagenic potential, was conducted using Salmonella typhimurium strains TA97, TA98, TA100, and TA102, with and without metabolic activation [8,27]. Much of the literature has demonstrated that many compounds that show positive responses in the Ames test are carcinogens in mammals. However, it must be noted that there is no absolute correlation between the two because they act through other nongenotoxic mechanisms or those absent in the bacterial cell [8,28]. With the activation of S9 and without S9.

Comparing the negative and positive control groups in the experimental group and ANOVA analysis, the liquid showed no mutagenic activity at all concentrations tested. Colony mutation data were not significantly increased, However, the number of reversed colonies in all the test strains increased significantly in the positive control in Table 1. The oral liquid formulation showed no significant induction of revertant colonies under either condition, confirming its lack of mutagenicity under the tested parameters.

Mammalian Bone Marrow Micronucleus Assay: the micronucleus test evaluates chromosomal integrity by quantifying micronuclei in polychromatic erythrocytes, which arise from chromosomal fragments or lagging chromosomes during cell division [29]. Mice were administered the oral liquid at 20 mL/kg/day. The micronucleus count (MCN ‰) in the treatment group (3.22 ± 1.03‰) showed no statistically significant difference from the negative control (2.44 ± 0.83‰, *p* > 0.05), whereas the positive control (cyclophosphamide, 40 mg/kg) exhibited a markedly elevated MCN ‰ of 88.23 ± 5.51‰ in Figure 1A. These results confirmed the absence of clastogenicity.

Sperm Morphology Assay: sperm morphology abnormalities were defined as amorphous, banana shape, folded, fat head, no hook, and double head/tail [30]. Sperm abnormalities were assessed in mice administered 20 mL/kg/day of the oral liquid. The sperm abnormality rate in the treatment group (1.21 ± 0.45%) was comparable to the negative control (1.38 ± 0.76%, *p* > 0.05) and significantly lower than the positive control (cyclophosphamide: 10.06 ± 13.87%, *p* < 0.01) in Figure 1B. As the negative control range (0.7 ± 3.4%) encompasses both control and treatment values, the oral liquid demonstrated no reproductive toxicity under these conditions.

#### 3.1.3. 28-Day Oral Toxicity Study

Hematological analysis is widely recognized as a sensitive indicator of drug and chemical toxicity in both human and animal models [22].

As summarized in Table 2, hematological parameters, reflecting the effect of the oral liquid on blood health, were measured and included red blood cell (RBC) count, white blood cell (WBC) count and differential, hemoglobin (HGB) concentration, hematocrit (HCT), and platelet (PLT) count. Through hematological analysis, as presented in Table 2, the results demonstrated that the hematological parameters Reacting the effect of oral liquid on blood health in mice. The Hemoglobin and Red Blood Cell Count related indicators reflect that 20 mL/kg/day oral solution did not have a significant effect on the red blood cells themselves and the oxygen supply capacity in the blood. White Blood Cell Count, Neutrophil Percentage, Lymphocyte Percentage, Monocyte Percentage, Eosinophil Percentage, and Basophil Percentage indicate that this dose of oral liquid does not produce severe immune response and hematological diseases. Moreover, in Platelet Count, the experimental group increased the platelet content, to some extent increasing hemostasis and coagulation function. Compared with the control group, red blood cells, white blood cells, hemoglobin and platelet count in the experimental group at a dose of 20 mL/kg/day exhibited no significant changes (*p* > 0.05). This indicates that the oral liquid at this dose has no adverse effects on hematopoietic function.

Meanwhile, serum biochemical analysis revealed that the Blood Urea Nitrogen/Creatinine index indicated no significant impact of the oral liquid on renal function. In terms of alanine aminotransferase, aspartate aminotransferase, total protein, albumin, and gamma globulin indices, it was found that the indices in the oral liquid group decreased significantly, suggesting that the oral liquid has a certain protective effect on liver function. The indices of total cholesterol and triglycerides also indirectly demonstrated that the oral liquid can play a role in weight loss and lipid lowering to a certain extent. Compared with the control group, the levels of aspartate aminotransferase, albumin, total cholesterol, and triglycerides in the experimental group of mice decreased significantly (*p* < 0.05), indicating a potential lipid regulating effect. In addition, the blood glucose of mice administered the oral liquid was significantly higher than that of the blank control group. Possible reasons include bioactive components in the formulation affect glucose metabolism by enhancing glycogenolysis or gluconeogenesis; or they promote lipolysis, with increased free fatty acids participating in hepatic glucose production through gluconeogenesis; it may also induce transient insulin resistance or regulate incretin responses. In this study, it may reflect enhanced metabolic mobilization to meet the energy demands of immune activation, which is consistent with the immunomodulatory effect of the oral liquid. However, its physiological significance and potential health impacts still need to be evaluated in future chronic toxicity and pharmacodynamic studies. Overall, the serum biochemical indices indicate that the oral liquid at this dose has no obvious hepatotoxicity or nephrotoxicity. Notwithstanding its compliance with OECD Guideline 407, the 28-day duration of this 28-day oral toxicity study assessment inherently precludes the evaluation of chronic effects. It is therefore imperative that subsequent chronic toxicity studies, aligned with OECD Guideline 451, be undertaken to establish a comprehensive safety profile for long-term clinical use.

### 3.2. Effects of Health Enhancing Oral Liquid on the Immune Function in Mice

In this study, the comprehensive effects of the oral liquid sample on the immune system of mice were systematically evaluated from four aspects: cellular immunity, humoral immunity, nonspecific immunity, and immune cell killing function, revealing its immunomodulatory properties and potential toxicological characteristics.

The spleen-to-body weight ratio is a standard gross indicator of immunomodulation. An increase can suggest immune stimulation, while a decrease might indicate immunosuppression. In terms of apparent results, as shown in Figure 2A, the oral liquid sample induced a dose dependent increase in the spleen/body weight ratio (*p* > 0.05). The oral liquid sample contained polysaccharide components that enhance immune activity. As the dose increases, these components gradually activate the immune system and stimulate lymphocyte proliferation, thus increasing the spleen/body weight ratio [10].

Cellular immunity deploys T cells to combat intracellular threats like infected or cancerous cells. Upon detecting foreign antigens, T cells activate, multiply, and specialize into helper T cells and cytotoxic T cells. This targeted defense concurrently establishes long-term immunological memory against recurrent threats [31]. In terms of cellular immunity, the oral liquid sample group showed a dose dependent promotion of lymphocyte proliferation. Although there was no statistical significance compared with the control group (*p* > 0.05) in Figure 2B, it enhanced the immune response to a certain extent. In Figure 2C,D, the results of the delayed type of hypersensitivity (DTH) test showed that the oral liquid could significantly inhibit the foot swelling induced by SRBC (*p* < 0.05; *p* < 0.01), indicating its regulatory effect on T cell mediated immune responses. In this experiment, after the intervention of the oral liquid, the foot swelling was significantly inhibited. It is highly likely that some active components in the oral liquid affected the activation process of T cells, hindered the recognition of antigens by T cells, or interfered with the proliferation, differentiation, and cytokine secretion of T cells, thus reducing the inflammatory response [10].

Humoral immunity is an adaptive immune process in which B lymphocytes recognize antigens, proliferate and differentiate into plasma cells with the assistance of helper T cells. Plasma cells secrete antibodies, which specifically bind to antigens and clear antigens by neutralizing toxins, blocking pathogen invasion, activating the complement system, and promoting phagocytosis [32]. In the humoral immunity experiment, both the serum hemolysin (HC50) level and the number of antibody producing cells increased significantly (*p* < 0.05; *p* < 0.01) in Figure 2E,F. The increase in the serum hemolysin level is a positive immune response of the body to invading antigens. More hemolysin can bind to and clear antigenic foreign substances such as pathogens, which helps maintain the internal environment stability and immune balance of the body and enhances the body’s resistance to infectious diseases [33]. The increase in the number of antibody producing cells is also conducive to the formation of immune memory. The generation of more memory B cells enables the body to initiate a faster and stronger immune response and produce a large number of antibodies when exposed to the same antigen again, thus improving the body’s long term immune defense ability against specific antigens [32].

Nonspecific immunity is an innate defense mechanism of the body. Through the physical barriers of the skin and mucous membranes, the phagocytosis of macrophages and other phagocytic cells, the activation of the complement system, and the immune regulation mediated by cytokines, it provides a non-targeted but rapidly initiated immune defense against a variety of pathogens [34]. The analysis of macrophage function showed that the oral liquid did not significantly affect the carbon clearance ability of monocytes/macrophages (*p* > 0.05) in Table 3, indicating that the oral liquid had no obvious promotion or inhibition effect on the basic function of macrophages in recognizing and clearing general nonspecific particles. The function of macrophages in this regard remained relatively stable, and its phagocytosis mechanism and related signal pathways did not change significantly due to the effect of the oral liquid. However, it significantly increased the phagocytosis of chicken red blood cells by peritoneal macrophages in a dose dependent manner (*p* < 0.01) in Table 3, which means that the phagocytosis ability of peritoneal macrophages to chicken red blood cells was continuously enhanced.

In addition, in terms of immune cell killing function, the sample increased the activity of NK cells (*p* < 0.05) in Figure 2G, but no clear dose–response relationship was observed. Immune cell killing function refers to the physiological process in which immune cells such as cytotoxic T cells and natural killer cells specifically or non-specifically recognize and clear pathogen infected cells, tumor cells, and other abnormal cells in the body through mechanisms such as releasing cytotoxic substances such as perforin and granzyme or activating the apoptosis signaling pathway to maintain the immune homeostasis of the body [34]. The polysaccharide components in the oral liquid can bind to specific receptors on the surface of NK cells, thereby triggering the activation signaling pathway of NK cells, enhancing the activity of NK cells [12].

Cellular immunity, humoral immunity, non-specific immunity, and immune cell killing function are interrelated and cooperate with each other in the immune system, jointly constituting a complete immune defense system. The experimental results show that the oral liquid sample affects the immunity of mice in many aspects, such as inducing a dose dependent increase in the spleen/body weight ratio, regulating cellular immunity and humoral immunity, affecting macrophage function, and enhancing NK cell activity. These results indicate that the components in the oral liquid sample, such as polysaccharides, may play an immunomodulatory role by activating immune cell signal pathways and regulating the proliferation and function of immune cells.

## 4. Conclusions

This study provides a comprehensive evaluation of an immune enhancing oral liquid formulated from PM and MO extracts. Acute toxicity studies showed an absence of abnormalities in mice and no treatment-related histopathological lesions in vital organs, indicating a favorable safety profile. Genotoxicity assessments—including the Ames test, mouse bone marrow micronucleus assay, and sperm morphology test—confirmed the non-mutagenicity of the formulation. 28-day oral toxicity study revealed no adverse effects on hematopoietic function; rather, the oral liquid demonstrated hepatoprotective effects and potential lipid-modulating properties. Immunological evaluations indicated that the formulation modulates both cellular and humoral immune responses, enhances macrophage phagocytic activity, and increases NK cell activity, collectively contributing to improved immune competence in mice.

Although the current findings provide foundational evidence supporting the immunomodulatory efficacy and general safety of the product, mechanistic insights into the active ingredients and their molecular targets remain limited. Addressing this gap should be a priority for future research, with emphasis on elucidating specific signaling pathways. Proposed mechanisms worthy of investigation include the TLR4/MyD88/NF-κB and MAPK pathways, which are central to macrophage activation and phagocytosis. The role of the JAK-STAT signaling pathway, particularly in NK cell activation and lymphocyte differentiation, also merits detailed analysis. Furthermore, the Nrf2/HO-1 axis represents a promising mechanistic target for explaining the observed hepatoprotective and antioxidant effects. Additionally, given the oral administration route, exploring the crosstalk among gut microbiota, microbial metabolites, and systemic immunity may yield novel insights into the immunomodulatory mechanisms. Finally, while this study assesses general and immunotoxicity, developmental and reproductive toxicity (DART) studies would be required for a complete safety assessment, particularly if this product were to be marketed for use by women of child-bearing age.

To complement these mechanistic studies, long-term toxicity experiments with extended dosing periods are imperative to comprehensively assess chronic exposure risks and the sustained influence on immune function. The integration of advanced omics technologies, such as transcriptomics and proteomics, will be pivotal for unbiased elucidation of the molecular regulatory networks modulated by the oral liquid. Expanding research into these areas will establish a more robust scientific foundation for optimizing the formulation and supporting its translational applications.

## Figures and Tables

**Figure 1 foods-14-03166-f001:**
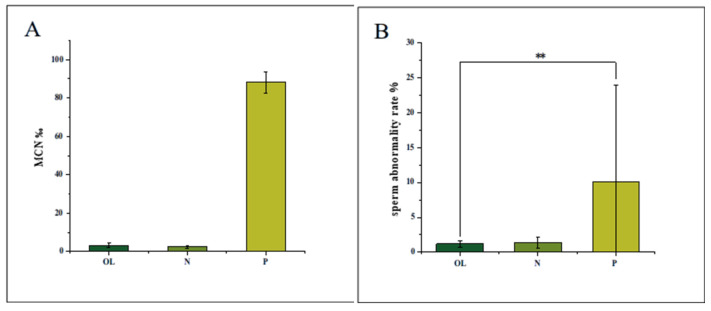
The results of genotoxicity assays; the result of mouse micronucleus test (**A**); the result of mouse sperm malformation experiments (**B**). Note: ** *p* < 0.01 is considered statistically significant. MCN ‰ = MNPCEs/PCEs × 1000‰; OL: Immune Enhancing Oral Liquid; N: distilled water; P: cyclophosphamide.

**Figure 2 foods-14-03166-f002:**
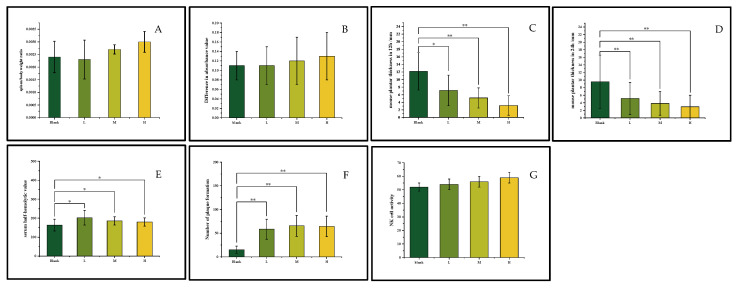
The results of effects on murine immunity; the spleen to body weight ratio (**A**); the result of lymphocyte proliferation (**B**); the result of delayed type hypersensitivity test in 12 h (**C**); the result of delayed type hypersensitivity test in 24 h (**D**); the result of serum hemolysin (**E**); the result of antibody producing cells (**F**); the result of activity of NK cells (**G**). Note: *compared with blank group; * *p* < 0.05 and ** *p* < 0.01 was considered statistically significant, L: low dose; M: medium dose; H: high dose.

**Table 1 foods-14-03166-t001:** Results of the bacterial reverse mutation assay.

Treatment	Dose (µg/Plate)	With S_9_				Without S_9_			
		TA_97_/num	TA_98_/num	TA_100_/num	TA_102_/num	TA_97_/num	TA_98_/num	TA_100_/num	TA_102_/num
Sample 1	5000	173.5 ± 9.5 ^b^	47.5 ± 7.5 ^b^	156.5 ± 4.5 ^b^	236.5 ± 11.5 ^b^	183.0 ± 11.0 ^b^	46.0 ± 6.0 ^b^	144.5 ± 5.5 ^b^	267.5 ± 10.5 ^b^
Sample 2	2500	174.5 ± 12.5 ^b^	42.0 ± 4.0 ^b^	135.0 ± 4.0 ^b^	245.5 ± 22.5 ^b^	179.5 ± 8.5 ^b^	37.5 ± 3.5 ^b^	149.5 ± 9.5 ^b^	272.0 ± 6.0 ^b^
Sample 3	1250	180.5 ± 6.5 ^b^	45.5 ± 2.5 ^b^	142.0 ± 5.0 ^b^	271.5 ± 14.5 ^b^	169.0 ± 9.0 ^b^	42.5 ± 2.5 ^b^	135.0 ± 17.0 ^b^	284.5 ± 10.5 ^b^
Sample 4	625	176.0 ± 14.0 ^b^	46.5 ± 2.5 ^b^	134.0 ± 10.0 ^b^	277.5 ± 16.5 ^b^	178.0 ± 9.0 ^b^	43.0 ± 3.0 ^b^	134.0 ± 10.0 ^b^	266.0 ± 9.0 ^b^
distilled water	5000	171.5 ± 9.5 ^b^	46.5 ± 3.5 ^b^	141.0 ± 10.0 ^b^	255.5 ± 15.5 ^b^	176.5 ± 11.5 ^b^	49.0 ± 4.0 ^b^	146.0 ± 11.0 ^b^	278.5 ± 6.5 ^b^
2-aminobacteria	50	1573.5 ± 95.5 ^a^	2123.0 ± 122.0 ^a^	1905.0 ± 87.0 ^a^	-	-	-	-	-
1,8 dihydroxyandone	50	-	-	-	755.5 ± 64.5 ^a^	-	-	-	-
dexon	1.5	-	-	-	-	1664.5 ± 143.5 ^a^	957.0 ± 132.0 ^a^	-	1016.5 ± 112.5 ^a^
sodium azide	10	-	-	-	-	-	-	2607.5 ± 101.5 ^a^	-

Note: Number of experimental repetitions (n = 6). In the same columns, different letters behind values represent they have significant difference (*p* < 0.05). The values are shown as the mean ± standard error.

**Table 2 foods-14-03166-t002:** Blood routine measures and biochemical measures of mice from the 28-day oral toxicity study.

Index	Blank Group	Sample
Hemoglobin (g/L)	139.00 ± 8.34	142.50 ± 12.65
Red Blood Cell Count (×10^12^/L)	9.35 ± 0.68	9.30 ± 1.04
White Blood Cell Count (×10^9^/L)	1.91 ± 0.75	1.95 ± 0.58
Neutrophil Percentage (%)	3.07 ± 1.22	2.77 ± 1.81
Absolute Neutrophil Count (×10^9^/L)	0.06 ± 0.07	0.05 ± 0.06
Lymphocyte Percentage (%)	85.46 ± 5.53	91.33 ± 10.21
Absolute Lymphocyte Count (×10^9^/L)	1.70 ± 0.59	1.69 ± 0.69
Monocyte Percentage (%)	4.01 ± 5.36	4.17 ± 3.26
Absolute Monocyte Count (×10^9^/L)	0.09 ± 0.15	0.11 ± 0.07
Eosinophil Percentage (%)	0	0
Absolute Eosinophil Count (×10^9^/L)	0	0
Basophil Percentage (%)	2.33 ± 1.94	2.53 ± 1.77
Absolute Basophil Count (×10^9^/L)	0.08 ± 0.07	0.09 ± 0.05
Platelet Count (×10^9^/L)	331.70 ± 265.5	405.50 ± 162.72
Glucose (GLU) mmol/L	6.58 ± 0.78	13.65 ± 1.66
Blood Urea Nitrogen (BUN) mmol/L	10.01 ± 0.94	11.64 ± 2.30
Creatinine (CREA) μmol/L	13.67 ± 1.35	15.32 ± 2.22
Alanine Aminotransferase (ALT) U/L	41.12 ± 6.54	47.52 ± 16.47
Aspartate Aminotransferase (AST) U/L	198.27 ± 38.75	101.25 ± 38.14 *
Total Protein (TP) g/L	59.85 ± 4.12	48.65 ± 3.02
Albumin (ALB) g/L	23.37 ± 2.10	16.39 ± 1.24 **
Gamma-Globulin (GLB) g/L	39.88 ± 3.45	32.79 ± 1.16 **
Total Cholesterol (TCHOL) mmol/L	3.54 ± 0.77	2.01 ± 0.25 **
Triglycerides (TG) mmol/L	3.31 ± 1.63	1.18 ± 0.33 **

Note: * *p* < 0.05 and ** *p* < 0.01 is considered statistically significant, * compared with blank group. Number of experimental repetitions (n = 10). The values are shown as the mean ± standard error.

**Table 3 foods-14-03166-t003:** Results of peritoneal macrophage phagocytosis.

Sample	Phagocytic Count K	Correct the Clearance Index of A	Percentage (%)	Phagocytic Count
blank	0.025 ± 0.0071	4.528 ± 0.74	2.69 ± 1.74	0.025 ± 0.020
L	0.021 ± 0.0081	4.133 ± 0.376	14.81 ± 5.66 **	0.203 ± 0.194 **
M	0.020 ± 0.0094	4.102 ± 0.454	20.79 ± 5.86 **	0.305 ± 0.106 **
H	0·020 ± 0·0065	4.307 ± 0.275	25.10 ± 5.41 **	0.330 ± 0.094 **

Note: compared with blank group; ** *p* < 0.01 is considered statistically significant. Number of experimental repetitions (n = 10). The values are shown as the mean ± standard error; L: low dose; M: medium dose; H: high dose.

## Data Availability

The original contributions presented in this study are included in the article, further inquiries can be directed to the corresponding author.

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
