# Peer review of "Comprehensive Evaluation of Toxicological Profile and Immunomodulatory Impact of an Immune Enhancing Oral Liquid"

_foods, 2025, doi:10.3390/foods14183166_

Round 1
Reviewer 1 Report
Comments and Suggestions for Authors
This manuscript presents a valuable toxicological and immunological evaluation of a herbal-based oral liquid formulation. The study design is generally sound, and the overall findings support the safety and potential immunomodulatory benefits of the formulation. However, the following points require clarification.
1. English language editing is strongly recommended.
Throughout the manuscript, there are numerous grammatical errors, awkward phrases, and unclear formulations that impede comprehension. For example:
- L.315–316: “Compared with other literature reports of PM liver toxic phenomenon…” is unclear and should be rephrased as “Compared with previously reported cases of liver toxicity associated with PM extracts…”
- L.343: “absence of clastogenic” → should be “absence of clastogenicity”
- L.441–442: “did not show a dose gradient effect” → more appropriate: “no clear dose-response relationship was observed”
- L.395–397: Sentence structure is overly complex and should be simplified.
A professional language review, ideally by a native English speaker or scientific editor, would significantly enhance clarity and precision.
2. Methods section requires more detail.
- The protocols for the micronucleus test (L.166–173) and sperm morphology assessment (L.174–184) are described too briefly. Please provide more information on slide preparation, scoring criteria, and how significance was determined.
- In the Ames test (Table 1), numerical data are provided, but it would be helpful to explicitly state whether all doses tested were within the acceptable baseline range for non-mutagenicity.
3. Toxicological findings: glucose elevation requires attention.
- As shown in Table 2 and discussed around L.380–382, a substantial increase in blood glucose levels was observed in the treated group (13.65 ± 1.66 mmol/L vs. 6.58 ± 0.78 mmol/L in controls). This change is not fully addressed in the discussion and could reflect a significant metabolic effect.
- A more thorough explanation is needed—e.g., potential mechanisms involving carbohydrate metabolism, or at least a note of caution regarding long-term metabolic implications.
4. Toxicity testing duration and limitations.
- The current subacute toxicity study is limited to 30 days (OECD 407). While this duration is standard for subacute testing, it does not allow conclusions about chronic safety for long-term human consumption.
- Please consider emphasizing this limitation more clearly in the Conclusion and suggesting the need for extended 90-day studies (OECD 408) or chronic exposure assessments.
5. Phytochemical composition and standardization.
- The supplementary data report the levels of sugars, flavonoids, and polyphenols. However, the reproducibility and batch-to-batch consistency of the product are not addressed.
- We recommend including a sentence in the discussion or conclusion acknowledging the importance of quality control and lot-to-lot chemical stability, especially for herbal mixtures.
6. Conclusion section could be more specific.
- The mention of future “omics” approaches (L.508–519) is appreciated but very general. Please consider suggesting specific molecular targets or immune pathways that might be explored in future work based on your current results.
Author Response
Thank you for your thorough and constructive review of our manuscript. We greatly appreciate the time and effort you have dedicated to providing these insightful comments, which have helped us significantly improve the quality of our work. We have carefully considered each point and revised the manuscript accordingly.

Reviewer 2 Report
Comments and Suggestions for Authors
The manuscript deals with an important topic that the food industry will face as it moves toward more natural ingredients, incorporating Traditional Eastern herbal products into US food ingredients or as new dietary ingredients. Primary issues with the manuscript: 1] need for input from a natural speaking English scientific colleague,2] detailed information in tabular formation of the concentrations of the ingredients in the products summing to 100%., 3] expressing units through out the paper as ug (mg)/kg/daily, and 4] specific but clear details for all methods.
ABSTRACT
Subacute = less than acute: OECD 407 is a repeat 28-day study.
"Non-toxic at the dose tested in the acute study
INTRODUCTION
Page 2, line 4 from bottom: "This proprietary composition," whereas elsewhere you indicated some of its composition. To repeat the study, others need the chemical details of the material tested.
Page 2: Suggest that DART studies need but have not been reported here? Also, no immunotoxicity assessment.
Page 3, first full paragraph, line 3. What does systematically mean?
MATERIALS AND METHODS
"The detailed experimental procedure was shown in Figure 1". This reviewer could not find Figure 1. [The figure labeled as Figure 1 deals with the micronucleus and mouse sperm malformation experiments.]
The first paragraph under 2.2 animal experiments is very confusing and must be clearly rewritten.
Section 2.3. Preparation and analysis of the Immune-Enhancing Oral Liquid.
Please clarify this paragraph. For example, "to ten pieces of what." A table showing the composition and concentration of the ingredients in this complex mixture should be added.
Section 2.4
A standard dose of 15 mL/kg/day = ug/mg/g/kg/day?
Units should be the same for all experiments.
Ames assay: mg or ug (table 1 says ug: please be consistent)
Dose for the mouse micronucleus and sperm maliformation experitments in mg/g/kg/day
Section 2.4.3 Should be 28 day repeat oral toxicity study not subacute
Units again! How was blood collected.
Control: water by gavage?
Please separate the Results and Discussion sections.
Acute toxicity test------what is the mean lethal dose? Insufficient doses to estimate one; no pathology was conducted only gross observations
top of page 8: are these the only important organs?
According to CPSC.........................................OK but what is the LD50 of this material?
3.1.3 28-day oral repeat toxicity study (units?) Single dose? Can only say "no effect" at the dose tested!
Much discussion in the Methods section
Figure 2E and 2F: serum hemolysin level (weak if at all) and antibody producing cells up (agree)
Page 9, line 7 from top of page: What dose?
Supplementary figure 1 and table 1- where are they referenced in the text?
Comments on the Quality of English Language
Needs work
Author Response

(The authors gave the same response as above.)

Reviewer 3 Report
Comments and Suggestions for Authors
Dear Authors,
I have carefully reviewed the manuscript, and while the study is generally interesting and comprehensive, I would like to provide a few suggestions and comments to further enhance its clarity and impact.
- Introduction
The introduction does a good job setting up the context for the study. It covers the potential benefits of the herbs, discusses safety concerns, and places the research within the regulatory framework. However, there are a few areas that could be improved to make it clearer and more informative, such as:
- It would be helpful to expand a bit on why PM and MO were chosen together. What makes them a good combination, and what is the expected synergy between the two?
- It would be useful to explain a bit more about why traditional water extraction were chosen, and how this affects the final product. Also, mentioning any quality control steps would add more depth.
- Please provide some details on how the polysaccharides and other compounds in PM and MO actually impact immune function at the molecular or cellular level.
2.1.
- The reagents used for each specific biochemical analysis (e.g., liver function tests, genotoxicity assays, immune function analyses, histopathology) should be listed separately. Each analysis must have its own set of reagents (like the manufacturer, catalog number etc.), to improve clarity and ensure reproducibility.
2.2.
- line 123- please specify the group for animal experiment (blank, sample…)
- line 124- Figure 1 (Line 463 is Figure 1. The results of genotoxicity assays) doesn’t match the text
2.3.
- please state the exact amount of the extract (mg) for the reproducibility
- The use of boiling water for extraction can lead to the degradation of sensitive bioactive compounds in the extracts (polyphenols, flavoniods, stilbenes etc.), potentially altering theit immunomodulatory effects. Since oral liquids, that enchance the immune system, are usually not prepared with boiling water, I recommend reconsidering this method or providing explanation for its use in order to better align it with standard practice
- line 136- add a brief description of the HPLC method and list the compounds that were determined using it (since this is also not stated in the figure/chromatogram shown in the results)
2.4.1.
- in the results section (line 305) LD50 is stated, but it has not been mentioned or described in this section. Please provide a detailed explanation of how the LD50 was determined
- please specify the dynamics of parameter measurement
2.4.2.
- Ames test: it is necessary to clarify what changes have been made in the protocol to make the methodology clarer
- Control doses: in the nucleus and sperm malformation tests, the coice of the doses and the test substances should be clarified in relation to standards in the literature
2.4.3.
- add reference to OECD guidelines: it is mentioned that guideline 187 (No. 407) was followed, but it should be clarified how the dose was chosen (especially in the context of the extracts used in this study)
- time of blood collection: blood samples were collected on day 31 after 30 days of treatment, but it should be clarified whether blood samples were takne before or after the last day of the treatment
2.5.
- dose selection: the rationale for dose selection should be explained- especially in relation to previous studies or dose to response data.
- spleen to body weight ratio: it would the useful to compare with a method or standard protocol for measuring spleen to body weight ratio and its significance in assessing immune function.
2.6.
- It is necessary to specify which type of ANOVA was used (e.g., one-way or two-way), and whether post-hoc tests were performed to further investigate significant differences.
- Additionally, it would be helpful to mention whether normality tests were conducted before applying parametric tests, and to clarify which correlation methods (if any) were use
3.1.1
- Please, add references to support your statements and to provide clearer comparison with the literature
- Clarify the method of LD50 determination (in the material and method section).
- Specify how observations were made during the 14-day period.
- Expand on the potential role of compatible detoxification in reducing toxicity, with relevant references.
3.1.2.
- reword the sentence “The oral liquid was found to be at any tested concentration (5000 μg/plate)” is unclear. It should specify whether the liquid showed no mutagenic activity at all concentrations tested.
- The comparison of negative and positive control groups should be described in more detail. It would be beneficial to clarify if there were any statistical tests applied to confirm the significant difference between the controls and experimental groups.
3.1.3.
- reword the sentence: “the hematological parameters Reacting the effect of oral liquid on blood health” is unclear.
- briefly explain what specific hematological parameters were measured (e.g., RBC count, WBC count, etc.) in a more concise way.
- Fix repetitions in the description of immune responses: There is some repetition in the descriptions of cellular immunity and humoral immunity. Streamlining these sections will improve clarity and readability.
- Clarify statistical methods: Although statistical significance (e.g., p < 0.05, p < 0.01) is mentioned, the specific statistical tests used to compare the groups should be stated. Additionally, further clarification of how the dose-dependent effect was determined (e.g., via ANOVA or regression analysis) would strengthen the interpretation of the results.
Author Response

(The authors gave the same response as above.)

Round 2
Reviewer 2 Report
Comments and Suggestions for Authors
The reviewer thanks the authors for their responses; however, the methods section is confusing, and the introduction lacks sufficient references to the newly added material.
Numerous words need to be italicized.
INTRODUCTION
page 2, line 65-71. needs appropriate references
page 2, line 74-76. needs an appropriate reference
page 2, lines 83-85. The sentence needs to be clarified.
page 3. Ministry of Health Drug Standard of the People's Republic of China [https://english.nmpa.gov.cn/2019-09/26/c_773012.htm]
page 3, lines 94-96. Reference needed
page 3. Please change subacute to 28-day oral toxicity study
METHODS
"were" need in several places in the opening paragraph
Total number of animals used and only males?
Overall, the methods section is incomplete and difficult to follow, making it difficult to understand the dosing and concentration of PM and MO in each experiment. For example, the 28-day study is reported to be OECD 407, but apparently, it was only a control and one dose, not three doses as per OECD 407. In one place, you state "following the 30-day treatment..........
On page 6, line 236, is this the same 28-day study or a different study? What is the sex of the mice?
Splenic Lymphocyte Proliferation Assay, Delayed-type hypersensitivity response, serum hemolysin assay,antibody-secreting cell quantification, carbon clearance kinetics, and the peritoneal macrophage phagocytosis assay (ten mice randomly selected) from the same group of mice were in vitro assays.
2.5.4. A new study---on the 2nd day after the last dose (28 or 30 days)
Comments on the Quality of English Language
Better
Author Response
我们衷心感谢审稿人抽出时间和宝贵的反馈。我们仔细考虑了提出的所有观点,并相应地修改了手稿。请在下面找到我们详细的逐点回复。手稿中的所有更改都使用“跟踪更改”功能突出显示。

Reviewer 3 Report
Comments and Suggestions for Authors
I appreciate the authors’ efforts in addressing my suggestions and revising the manuscript accordingly.
Author Response
Dear Reviewer 3,
Thank you for your email and the positive feedback on our revisions. We are delighted to hear that the reviewers and you are satisfied with our responses and the modifications made to the manuscript.
We look forward to the next steps in the publication process.
Sincerely,
Kan Qixin
On behalf of all co-authors.